

# Hybridizing sequential and variational data assimilation for robust high-resolution hydrologic forecasting

Felipe Hernández, Xu Liang

Civil and Environmental Engineering Department, University of Pittsburgh, Pittsburgh, 15213, United States of America

*Correspondence to*: Xu Liang (xuliang@pitt.edu)

**Abstract.** There are two main frameworks for the estimation of initial states in geophysical models for real-time and forecasting applications: sequential data assimilation and variational data assimilation. However, modern high-resolution models offer challenges, both in terms of indeterminacy and computational requirements, which render most traditional methods insufficient. In this article we introduce a hybrid algorithm called OPTIMISTS which combines advantageous

features from both of these data assimilation perspectives. These features are integrated with a multi-objective approach for selecting ensemble members to create a probabilistic estimate of the state variables, which promotes the reduction of observational errors as well as the maintenance of the dynamic consistency of states. Additionally, we propose simplified computations as alternatives aimed at reducing memory and processor requirements. OPTIMISTS was tested on two models of real watersheds, one with over 1,000 variables and the second with over 30,000, on two distributed hydrologic modelling

engines: VIC and the DHSVM. Our tests, consisting of assimilating streamflow observations, allowed determining which features of the traditional approaches lead to more accurate forecasts while at the same time making an efficient use of the available computational resources. The results also demonstrated the benefits of the coupled probabilistic/multi-objective approach, which proved instrumental in reducing the harmful effects of overfitting—especially on the model with higher dimensionality.

**1 Introduction**

Despite increasing availability of Earth-sensing data, geophysical models are as underdetermined as ever because of their growing complexity. Taking advantage of distributed physics and the mounting availability of computational power, these models have the potential to more accurately represent geophysical processes. This is achieved through the inclusion of numerous, usually unknown parameters and state variables, but at the cost of abandoning parsimony. In order to be able to

25 rely on these high-resolution models for critical real-time and forecast applications, considerable improvements on traditional parameter and initial state estimation techniques must be made with two main goals. First, to allow for an efficient management of the huge number of unknowns. And second, to mitigate the harmful effects of overfitting—i.e., the loss of forecast skill due to an over-reliance on the calibration/training data (Hawkins, 2004). Because of their numerous degrees of





freedom, overfitting is a much bigger threat in high-dimensional models due to the phenomenon of equifinality (Beven, 2006).

There are two main "schools" for the estimation of a model's initial state variables through the incorporation of available observations: sequential data assimilation, which follows Bayesian theory; and variational data assimilation, which takes an

optimization perspective. Embracing the unavoidability of equifinality, sequential data assimilation creates probabilistic estimates of the state variables in an attempt to also capture their uncertainty. These state probability distributions are adjusted every time step to better match the observations using Bayes' theorem. While the seminal Kalman Filter (KF) is constrained to linear dynamics and Gaussian distributions, Ensemble Kalman Filters (EnKF) can support non-linear models (Evensen, 2009), and Particle Filters (PF) can also manage non-Gaussian estimates for added accuracy (Smith et al., 2013).

Recent research has focused on efforts to improve the efficiency of EnKFs (Li et al., 2015) and PFs (van Leeuwen, 2015) to make them applicable to high-resolution models.

On the other hand, variational data assimilation is more akin to traditional calibration approaches (Efstratiadis and Koutsoyiannis, 2010). It seeks to find a single/deterministic initial state variable combination that minimizes the departures (or "variations") of the modelled values from the observations throughout an analysis window (Reichle et al., 2001). If the

model's dynamics are linearized, the optimum initial state can be very efficiently found in the resulting convex search space by using gradient methods. While this feature has made 4-dimensional variational (4DVar) variants very popular in meteorology and oceanography (Ghil and Malanotte-Rizzoli, 1991), their application in hydrology has been less widespread because of the difficulty of linearizing land-surface physics (Liu and Gupta, 2007). Moreover, variational data assimilation requires the inclusion of computationally expensive adjoint models if one wishes to account for the state estimates'

uncertainty (Errico, 1997).

Both perspectives on data assimilation have interesting characteristics and there have been multiple attempts at combining them in the recent literature. For example, sequential filters have been used as adjoints in 4DVar to enable probabilistic estimates (Zhang et al., 2009). Similarly, weak-constrained 4DVar allows state estimates to be determined at several time steps within the assimilation time window and not only at the beginning (Trémolet, 2006). However, this comes at the cost

of having to determine the state variables during the entire assimilation window, considerably increasing the dimensionality of the problem. On the opposite side, some sequential filters have been coupled with optimization approaches to select ensemble members (Dumedah and Coulibaly, 2013), or have extended the analysis of candidate state estimates to multiple time-step spans (Noh et al., 2011). 4DEnVar (Buehner et al., 2010), a fully-hybridized algorithm, is gaining increased attention for weather prediction (Desroziers et al., 2014; Lorenc et al., 2015). However, we believe that this hybrid still

inherits some disadvantages from its parent algorithms that can be improved upon.

In this article we introduce OPTIMISTS (for Optimized PareTo Inverse Modelling through Integrated STochastic Search), a hybrid data assimilation algorithm that incorporates the features that we consider most valuable from both sequential and variational methods. The choice of these features and the design of their interactions were guided by the two stated goals: to allow for practical scalability to high-dimensional models, and to enable balancing the imperfect observations and the





imperfect model estimates to minimize overfitting. Table 1 summarizes the main characteristics of standard sequential and variational approaches and their contrasts with OPTIMISTS.

## 2 Proposed data assimilation algorithm

OPTIMISTS, our proposed algorithm, allows selecting a flexible data assimilation time step $\Delta t$—i.e., the time window at
which candidate state configurations are compared to observations. It can be as small as the model or the observation time step—as in sequential approaches—or as big as the entire assimilation window—as in variational ones. For each assimilation time step at time $t$ a new state probability distribution $\boldsymbol{S}^{t+\Delta t}$ is estimated from the current distribution $\boldsymbol{S}^t$, the model, and one or more observations $\boldsymbol{o}_{\text{obs}}^{t:t+\Delta t}$. The distributions are determined from a set of weighted "root" or "base" sample states $\boldsymbol{s}_i$ using multivariate weighted kernel density estimation (West, 1993). Each of the samples or ensemble members $\boldsymbol{s}_i$ is comprised of
a value vector for the state variables. The objective of the algorithm is then to produce a set of $n$ samples $\boldsymbol{s}_i^{t+\Delta t}$ with corresponding weights $w_i$ for the next assimilation time step to determine the target distribution $\boldsymbol{S}^{t+\Delta t}$. This process is repeated iteratively each assimilation step until the entire assimilation time frame is covered, at which point the resulting distribution can be used to perform the forecast simulations. Here OPTIMISTS' main loop is described, while the details regarding the state probability distributions are explained in Sect. 2.1.

Let a "particle" $\boldsymbol{P}_i$ be defined by a source state $\boldsymbol{s}_i^t$, a corresponding target state $\boldsymbol{s}_i^{t+\Delta t}$, a set of output values $\boldsymbol{o}_i^{t:t+\Delta t}$, a set of fitness metrics $\boldsymbol{f}_i$, a rank $r_i$, and a weight $w_i$. (The denomination of particle stems from the PF literature, and is analogous to the "member" term in EnKFs.) The fitness metrics are used to compare particles with each other in the light of one or more optimization objectives. The main loop consists of the following steps which are further explained below. Table 2 lists the seven global parameters of the algorithm.

1.   Drawing: draw root samples $\boldsymbol{s}_i^t$ from $\boldsymbol{S}^t$ in descending weight order until $\sum w_i \geq w_{\text{root}}$

2.   Sampling: randomly sample $\boldsymbol{S}^t$ until the total number of samples in the ensemble is $p_{\text{samp}} * n$

3.   Simulation: compute $\boldsymbol{s}_i^{t+\Delta t}$ and $\boldsymbol{o}_i^{t:t+\Delta t}$ from each non-evaluated sample $\boldsymbol{s}_i^t$ using the model

4.   Evaluation: compute the fitness values $\boldsymbol{f}_i$ for each sample/particle

5.   Optimization: create additional samples using evolutionary algorithms and return to 3 (if number of samples is below $n$)

6.   Ranking: assign ranks $r_i$ to all particles $\boldsymbol{P}_i$ using non-dominated sorting

7.   Weighting: compute the weight $w_i$ for each particle $\boldsymbol{P}_i$ based on its rank $r_i$

While traditional PFs draw all the root samples from $\boldsymbol{S}^t$ (Gordon et al., 1993), OPTIMISTS only draws a subset of them and then completes the ensemble with random samples and/or samples created by the optimization algorithm. The distinction
between root samples and random samples is that the former are those that define the probability distribution $\boldsymbol{S}^t$ (that serve as centroids for the kernels), while the latter are generated stochastically from the kernels. $w_{\text{root}}$ represents the minimum





fraction of the total weight of the root samples that are to be drawn in the drawing step. The root samples with the highest weight—those which are the "best performers"—should be drawn first. Once the total weight of the drawn samples $\sum w_i$ reaches $w_{\text{root}}$, we proceed to the sampling step. This second step should contribute to the diversity of the ensemble in order to avoid sample impoverishment as seen on PFs (Carpenter et al., 1999), and serve as a replacement for traditional

resampling (Liu and Chen, 1998). The algorithm then uses the model to compute the resulting state vector $\boldsymbol{s}_i^{t+\Delta t}$ and an additional set of output variables $\boldsymbol{o}_i^{t:t+\Delta t}$ for each of the samples.

In the evaluation step, OPTIMISTS computes the fitness metrics $\boldsymbol{f}_i$ using user-defined fitness functions, which should allow one to determine if any two particles are equally desirable or if one of them is preferred over the other. Users may define observational error functions to be minimized or observational likelihood functions to be maximized—if the uncertainty of

10 the observations is known. One of the main features of the algorithm is its ability to judge candidate particles in light of multiple criteria simultaneously. Therefore, if multiple observations are available (Montzka et al., 2012), several objectives can be thus defined. Moreover, we consider that including an additional objective which favours particles that are more consistent with previous time steps, and thus enable better enforcing of the mass and energy conservation laws, would ultimately contribute to reduce overfitting by counterbalancing the traditional single-minded focus on observations. An array

of measures of consistency, from very simple (Dumedah et al., 2011) to very complex (Ning et al., 2014) ones, has been proposed. Here we suggest taking advantage of the probabilistic approach by attempting to maximize the likelihood of source states $\boldsymbol{s}_i^t$ given the current distribution $\boldsymbol{S}^t$. Even when consistency terms are used in single-objective approaches by lumping them into a unique cost function, we expect multi-objective strategies to lead to a better balance.

OPTIMISTS features an ensemble multi-objective global optimization algorithm similar to AMALGAM (Vrugt and

20 Robinson, 2007), but that allows model simulations to be run in parallel (Crainic and Toulouse, 2010). The optimization framework includes a genetic algorithm (Deb et al., 2002) and a hybrid approach that combines ant colony optimization (Socha and Dorigo, 2008) and Metropolis-Hastings sampling (Haario et al., 2001). These evolutionary algorithms can efficiently explore non-convex spaces thus relieving the need to linearize the model's dynamics as is done in 4DVar. The optimization algorithm creates additional samples $\boldsymbol{s}_i^t$ based on the ones already in the ensemble, in batches of size $n_{\text{pop}}$, and

25 stops when the total number of samples reaches $n$. The simulation and evaluation steps are repeated iteratively for each one of these new batches of samples. The $p_{\text{samp}}$ parameter defines what fraction of the total samples $n$ are to be created through sampling (draws or random); the complement, $\left(1 - p_{\text{samp}}\right) \cdot n$, indicates how many are to be created using the optimization approach.

In the ranking step we employ non-dominated sorting (Deb, 2014), another successful technique from the multi-objective

optimization literature, in order to balance the potential tensions between the various objectives. This sorting criteria is based on the concept of "dominance" instead of organizing all particles from the "best" to the "worst". A particle dominates another if it outperforms it according to at least one of the criteria/objectives while simultaneously is not outperformed according to any of the others. Following this principle, particles can be grouped in "fronts" comprised of members which



are mutually non-dominated; that is, none of them is dominated by any of the rest. Particles in a front, therefore, represent the effective trade-offs between the competing criteria. Figure 1.a illustrates the result of non-dominated sorting applied to nine particles being analysed under the minimum observational error and maximum source state likelihood objectives. In our implementation we use the fast non-dominated sorting algorithm to define the fronts and assign the corresponding ranks $r_i$

(Deb et al., 2002). More efficient non-dominated sorting alternatives are available if performance becomes an issue (Zhang et al., 2015).

In the final step, OPTIMISTS assigns weights $w_i$ to each particle based on its rank $r_i$ as shown in Eqs. (1) and (2). This Gaussian weighting is based on the ensemble size $n$ and the greed parameter $g$, and is similar to that proposed by (Socha and Dorigo, 2008). When $g$ is equal to zero, particles in all fronts are weighted uniformly; when $g$ is equal to one, only particles

in the Pareto/first front are assigned non-zero weights. With this, the final estimated probability distribution of state variables for the next time step $\boldsymbol{S}^{t+\Delta t}$ can be established using multivariate weighted kernel density estimation as demonstrated in Fig. 1.b.

$$w_i = \frac{1}{\sigma\sqrt{2\pi}} e^{-\frac{(r_i-1)^2}{2\sigma^2}} \tag{1}$$

$$\sigma = n \cdot [0.1 + 9.9 \cdot (1-g)^5] \tag{2}$$

**2.1 Model state probability distributions**

OPTIMISTS requires two computations related to the state-variable probability distribution $\boldsymbol{S}^t$: obtaining the likelihood of a

sample (for the evaluation step) and generating random samples (for the sampling step). The probability density of a weighted kernel density distribution $\boldsymbol{S}^t$ at a given point (here we use the density interchangeably with the likelihood function $\mathcal{L}$) can be computed using Eq. (3) (Wand and Jones, 1994). If we use Gaussian kernels, the kernel function $K$, parameterized by the bandwidth matrix $\mathbf{B}$, is evaluated using Eq. (4).

$$\mathcal{L}(\boldsymbol{s}|\boldsymbol{S}) = \frac{1}{\sum w_i} \sum_{i=1}^{n} [w_i \cdot K_{\mathbf{B}}(\boldsymbol{s} - \boldsymbol{s}_i)] \tag{3}$$

$$K_{\mathbf{B}}^{\text{Gauss}}(\boldsymbol{z}) = \frac{1}{\sqrt{(2\pi)^n \cdot |\mathbf{B}|}} \exp\left(-\frac{1}{2}\boldsymbol{z}^{\text{T}}\mathbf{B}^{-1}\boldsymbol{z}\right) \tag{4}$$

Matrix $\mathbf{B}$ is the covariance matrix of the kernels, and thus determines their spread and orientation in the state space. $\mathbf{B}$ is of

size $d \times d$, where $d$ is the dimensionality of the state distribution (i.e., the number of variables). Several optimization-based methods exist to compute $\mathbf{B}$ by attempting to minimize the asymptotic mean integrated squared error (AMISE) (Duong and Hazelton, 2005; Sheather and Jones, 1991). However, here we opt to use a simplified approach for the sake of computational efficiency: we determine $\mathbf{B}$ by scaling down the sample covariance matrix $\mathbf{C}$ using Silverman's rule of thumb, which takes into account the number of samples $n$ and the dimensionality of the distribution $d$, as shown in Eq. (5) (Silverman, 1986).

Figure 1.b shows the density of a two-dimensional example distribution using this method. If computational constraints are





not a concern, using AMISE-based methods or kernels with variable bandwidth (Hazelton, 2003; Terrell and Scott, 1992) can result in higher accuracy.

$$\mathbf{B}^{\text{Silverman}} = \left(\frac{4}{d+2}\right)^{\frac{2}{d+4}} \cdot n^{-\frac{2}{d+4}} \cdot \mathbf{C} \tag{5}$$

Secondly, we can generate random samples from a multivariate weighted kernel density distribution by dividing the problem into two: we first select the root sample and then generate a random sample from the kernel associated with that base sample. The first step corresponds to randomly sampling a multinomial distribution with $n$ categories and assigning the normalized weights of the particles as the probability of each category. Once we select a root sample $s_{\text{root}}$, we can generate a random sample $s_{\text{random}}$ from a vector $v$ of independent standard normal random values of size $d$, and a matrix $\mathbf{A}$ as shown in Eq. (6). If $\mathbf{B}$ is positive-definite, $\mathbf{A}$ can be computed from a Cholesky decomposition (Krishnamoorthy and Menon, 2011) such that $\mathbf{A}\mathbf{A}^{\text{T}} = \mathbf{B}$. Otherwise, an eigendecomposition can be used to obtain $\mathbf{Q}\mathbf{\Lambda}\mathbf{Q}^{\text{T}} = \mathbf{B}$ to then set $\mathbf{A} = \mathbf{Q}\mathbf{\Lambda}^{\frac{1}{2}}$.

$$s_{\text{random}} = s_{\text{root}} + \mathbf{A}v \tag{6}$$

Both computations (likelihood and sampling) require that $\mathbf{B}$ is invertible and, therefore, that none of the variables have zero variance or are perfectly linearly dependent on each other. Zero-variance variables must therefore be isolated and $\mathbf{B}$ marginalized before attempting to use Eq. (4) or to compute $\mathbf{A}$. Similarly, linear dependences must also be identified beforehand. If we include variables one by one, we can determine if a newly added one is linearly dependent if the determinant of the extended sample covariance matrix $\mathbf{C}$ is zero. Once identified, we can efficiently compute the regression coefficients for the dependent variable from $\mathbf{C}$ following the method described by (Friedman et al., 2008). The constant coefficient of the regression must also be calculated for future reference. For each dependent variable we thus determine a linear model that is represented by a set of regression coefficients.

## 2.2 High-dimensional state vectors

When the state vector of the model becomes larger (i.e., $d$ increases), as is the case for distributed high-resolution numerical models, difficulties start to arise when dealing with the computations involving the probability distribution. At first, the likelihood, as computed with Eqs. (3) and (4), tends to diverge either towards zero or towards infinity. This phenomenon is related to the normalization of the density—so that it can integrate to one—and to its fast exponential decay as a function of the sample's distance from the kernel's centres. In these cases we propose using an approximated formulation that makes the density proportional to the inverse square Mahalanobis distance (Mahalanobis, 1936) to the root samples, thus skipping the exponentiation and normalization operations of the Gaussian density. This simplification, which would correspond to the inverse square difference between the sample value and the kernel's mean in the univariate case, is shown in Eq. (7). The resulting distortion of the Gaussian bell-curve shape should not affect the results significantly, as OPTIMISTS uses the fitness functions only to check for domination between particles—so only the relative differences between likelihood values are important and not their actual magnitudes.



$$\mathcal{L}(s|S) \approx \mathcal{L}^{\text{Mahalanobis}}(s|S) = \frac{1}{\sum w_i} \sum_{i=1}^{n} \frac{w_i}{|(s - s_i)^{\mathrm{T}} \mathbf{B}^{-1}(s - s_i)|} \tag{7}$$

However, computational constraints might also make this simplified approach unfeasible both due to the $O(d^2)$ space requirements for storing the bandwidth matrix **B** and the $O(d^3)$ time complexity of the decomposition algorithms, which rapidly become huge burdens for the memory and the processors. Therefore, we can chose to sacrifice some accuracy by using a diagonal bandwidth matrix **B** which does not include any covariance term—only the variance terms in the diagonal

are computed and stored. This means that, even though the multiplicity of root samples would help in maintaining a large portion of the covariance, another portion is lost by preventing the kernels from reflecting the existing correlations. In other words, variables would not be rendered completely independent, but rather conditionally independent because the kernels are still centred on the set of root samples. Kernels using diagonal bandwidth matrices are referred to as D-class while those using the full covariance matrix are referred to as F-class. The $k_{\text{F−class}}$ parameter controls which case is used in

OPTIMISTS.

With only the diagonal terms of matrix **B** available ($b_{jj}$), we opt to roughly approximate the likelihood by computing the average of the standardized marginal likelihood value for each variable $j$, as shown in Eq. (8). Independent/marginal random sampling of each variable can also be applied to replace Eq. (6) by adding random Gaussian residuals to the elements of the selected root sample $s_{\text{root}}$. Sparse bandwidth matrices (Friedman et al., 2008; Ghil and Malanotte-Rizzoli, 1991) or low-rank

approximations (Ghorbanidehno et al., 2015) could be worthwhile intermediate alternatives to our proposed quasi-independent approach, to be explored in the future.

$$\mathcal{L}(s|S) \approx \mathcal{L}^{\text{independent}}(s|S) = \frac{1}{d\sqrt{2\pi} \sum w_i} \sum_{j=1}^{d} \sum_{i=1}^{n} \left\{ w_i \cdot \exp\left[ -\frac{(s_j - s_{i,j})^2}{2b_{jj}} \right] \right\} \tag{8}$$

## 3 Case studies

We now investigate whether if the use of OPTIMISTS can help improve the forecasting skill of hydrologic models. Is the algorithm suitable for high-resolution models? Additionally, which parameters should one chose to obtain better results? To

help address these questions, we coupled a Java implementation of OPTIMISTS with two popular open-source distributed hydrologic modelling frameworks: Variable Infiltration Capacity (VIC) (Liang and Xie, 2001, 2003; Liang et al., 1994, 1996a, 1996b) and the Distributed Hydrology Soil and Vegetation Model (DHSVM) (Wigmosta et al., 2002, 1994). The DHSVM was conceived for high-resolution representations of the Earth's surface, allowing for 3D saturated and unsaturated subsurface flow routing and 1D/2D surface flow routing. VIC, on the other hand, is targeted at much larger watersheds

focusing on vertical subsurface dynamics, but enabling intra-cell soil and vegetation heterogeneity. Both engines needed several modifications so that they could be executed in a non-continuous fashion as required for data assimilation. More specifically, we modified a version of VIC, enhanced with overland routing capabilities (Wen et al., 2012), to be able to save



and open routing state information; and fixed several important bugs in version 3.2.1 of the DHSVM mostly related to the initialization of state variables, but also pertaining to routing data and physics.

We selected two distributed models to perform streamflow forecasting tests using OPTIMISTS: a coarse-resolution VIC model of the Blue River watershed in Oklahoma, and a high-resolution DHSVM model of the Indiantown Run watershed in Pennsylvania. Table 3 shows a list of the main characteristics of the two test watersheds and their associated models. Figure 2 shows their land cover map together with the layout of the modelling cells. Eight parameters of the Blue River model were calibrated, using a Shuffled Complex Evolution (SCE) algorithm that is coupled with VIC (Parada et al., 2003), so as to minimize the traditional $\ell_2$-norm Nash-Sutcliffe Efficiency ($\text{NSE}_{\ell_2}$) coefficient (Nash and Sutcliffe, 1970) measured with respect to daily streamflow data from the corresponding USGS station. A parameter combination was selected, out of 10,000 that were evaluated, that yielded a $\text{NSE}_{\ell_2}$ of 0.63. In contrast, the multi-objective ensemble optimization algorithm used on OPTIMISTS was employed to calibrate 18 parameters of the Indiantown Run model using hourly streamflow data. Three error metrics were optimized simultaneously: the $\text{NSE}_{\ell_2}$, the mean absolute relative error (MARE), and the absolute bias. Out of 2,575 evaluated candidates, we chose a parameterization with $\text{NSE}_{\ell_2} = 0.81$, MARE = 37.85%, and an absolute bias of 11.83 l/s.

These "optimal" parameter sets, together with additional sets produced in the optimization process were used to run the models and determine different state variable vectors $\boldsymbol{s}$ to produce the state probability distributions at the beginning of each data assimilation scenario $\boldsymbol{S}^0$. The state variables include liquid and solid interception; ponding, water equivalent and temperature of the snow packs; and moisture and temperature of each of the soil layers. The Blue River model has 20 cells, with a maximum of seven intra-cell soil/vegetation partitions. Adding the stream network variables, the model has a total of $d = 1,105$ state variables. The Indiantown Run model has a total of 1,472 cells and $d = 33,455$ state variables.

To assess the performance of OPTIMISTS, three forecasting scenarios (Scenario 1, 2, and 3) were selected for the Blue River and two (Scenario 1 and 2) for the Indiantown Run, each of them with different levels of fit between the calibrated model's simulations and the observed hydrographs. We used factorial experiments (Montgomery, 2012) to test different configurations of OPTIMISTS on each of these scenarios, by first assimilating the streamflow data for two or four weeks, and then measuring the forecasting performance for the subsequent two weeks. The forecast was produced by running the model using each of the samples $\boldsymbol{s}_i$ from the state distribution $\boldsymbol{S}$ at the end of the assimilation time period, and then using the samples' weights $w_i$ to produce an average forecast. Deterministic model parameters (those from the calibrated models) and forcings were used in all simulations.

Table 4 shows the setup of each of the three full factorial experiments we conducted showing the selected set of assignments for OPTIMISTS' parameters. The mean absolute error (MAE) with respect to the streamflow observations was used as an objective in all cases. For the Blue River scenarios, the likelihood objective (when used) was computed using either Eq. (8) if $k_{\text{F-class}}$ was set to false, or Eq. (7) if $k_{\text{F-class}}$ was set to true. Equation (8) was used for all Indiantown Run scenarios. The assimilation period was of two weeks for most configurations, except for those in Experiment 3 with $\Delta t = 4$ weeks. During





both the assimilation and the forecasting periods we used unaltered streamflow data from the USGS and forcing data from NLDAS-2 (Cosgrove et al., 2003)—even though a forecasted forcing would be used instead in an operational setting (e.g., from systems like NAM (Rogers et al., 2009) or ECMWF (Molteni et al., 1996)).

## 3 Results and discussion

From the forecasts obtained in the three factorial experiments, we calculated three performance measures for the streamflow ensemble mean with respect to the observations: the $NSE_{\ell_2}$ (which focuses mostly on the peaks of hydrographs), an $\ell_1$-norm version of the Nash-Sutcliffe Efficiency coefficient ($NSE_{\ell_1}$) (Krause et al., 2005), and the MARE (which focuses mostly on the valleys). Figure 3 shows the streamflow comparison between the observations, the default model (the calibrated model without undergoing data assimilation), and the ensemble mean of a probabilistic simulation using a good-performing

configuration of OPTIMISTS for the three Blue River scenarios (those corresponding to Experiment 1). We can see that the default model has a tendency to overestimate the low flow and that OPTIMISTS can contribute to the improvement of the estimation of these valleys. However, it seems that, by itself, it cannot compensate completely for what possibly are structural and parametrical errors in the model. The supplementary material includes the performance metrics for all experiments and scenarios, both for their assimilation periods and their forecast periods.

Figure 4 summarizes how much the OPTIMISTS simulations improved over the forecasts from the default model by marginalizing the results for each scenario or parameter assignment in Experiment 1 (i.e., each column of the boxplot shows the distribution of all configurations of the specified scenario or the specified parameter assignment). When using data assimilation, the performance is reduced in most cases for Scenario 1, as measured by the $NSE_{\ell_2}$ and the $NSE_{\ell_1}$—but not by the MARE. Scenario 2 displays a mixed performance for these two metrics, but the forecast is always improved for Scenario

3. While the poor results in the first scenario can be attributed in large part to the fact that the default model is already very close to the observations, and thus leaves little room for improvement, it is worrisome that using data assimilation would in fact impoverish estimations. Similarly, the experiment showed that reducing the error during the assimilation period does not necessarily produce a better forecast: a common thread was that some configurations would raise the soil moisture in an attempt to mimic the peak flow values, but would later fail to return to the low regime of the recession curve due to the

previously-induced wetness. For Experiment 1, the correlation between the improvement of the $NSE_{\ell_2}$ during the assimilation period and the improvement during the forecast period was of -0.344, and of -0.669 for the $NSE_{\ell_1}$. While there was a positive correlation for the reduction of the MARE of 0.631, these numbers clearly show that the model is very highly prone to overfitting when performing data assimilation, and that focusing only on reducing the error during the training period might indeed be a misleading strategy.

The results shown in Figure 4 can help us determine which of the strategies incorporated into OPTIMISTS are better at improving the forecasts. We can see that the errors are considerably reduced when using F-class kernels. However, there is not enough statistically-significant evidence that any of the other parameters had an effect on the results, as evidenced by the



$p$-values from the analysis of variance (ANOVA)—the supplementary material includes the ANOVA tables for all experiments. This means that using the secondary likelihood objective, increasing the size of the ensemble $n$, changing the assimilation time step $\Delta t$, or complementing Bayesian sampling with optimization algorithms do not meaningfully contribute to improving the forecasts. It is especially surprising that the performance is virtually indifferent to $n$, as it is widely held in

the data assimilation literature that particle-based methods are only as powerful as the size of their ensembles (Snyder et al., 2008). If these results cannot be generalized to challenge this entrenched assumption, they at least indicate that OPTIMISTS can efficiently encode the state probability distribution of complex models using relatively few particles—which directly translates to better scalability to higher-dimension applications.

Figure 5 shows the hydrograph comparisons for scenarios 1 and 2 for the Indiantown Run (experiments 2 and 3). In both

cases we see the default model becoming too dry throughout, possibly because of recessions occurring faster than they should, at least during the period being studied. Assimilating streamflow data with OPTIMISTS can lead to improvements in both cases. Figure 6 summarizes the improvements achieved in the configurations of Experiment 2. For Scenario 1 we can see a general increase of the Nash-Sutcliffe coefficients, but a corresponding decline in the MARE, evidencing tension between getting the peaks or the valleys right. For both scenarios there are configurations that performed very poorly, and

we can look at the marginalized results in the boxplots for clues into which parameters might have caused this.

In the first place, there is strong statistical evidence to conclude that configurations with an assimilation time step $\Delta t$ of the entire two weeks considerably outperform those using a 1-hour time step (the former almost always improved the forecasts). We attribute this to the ability of the larger time window to provide a better grasp of the entire hydrograph; while the sequential approach probably leads to overfitting by focusing on short time frames one at a time. Similarly, we can see that

adding the likelihood objective considerably improves the results. Not only does it seem that for this model the secondary objective would mitigate the effects of overfitting, but it was interesting to note some configurations in which using it would actually help in achieving a better fit during the assimilation period. This provides further evidence in favour of our initial hypothesis that imposing a likelihood standard to candidate particles does indeed help maintaining the temporary consistency of the model's dynamics. The reason for the secondary objective not being advantageous in the Blue River model, on the

other hand, is consistent with the hypothesis under which higher dimensionality leads to more significant equifinality problems (although we did not find the same negative correlations between improving the fit during the assimilation and forecasting periods in the Indiantown Run).

While the ANOVA also provided evidence against the use of optimization algorithms, we are reluctant to instantly rule them out on the grounds that there were statistically significant interactions with other parameters. The optimizers led to poor

results in cases with 1-hour time steps or when only the error objective was used, again supporting the general idea that excessive focus on reducing the error, especially on a limited analysis window, can lead to overfitting. Other statistically significant results point to the benefits of using the root samples more intensively (in opposition to using random samples) and, to a lesser extent, to the benefits of maintaining an ensemble of moderate size.



Finally, Fig. 7 shows the summarized improvements from Experiment 3. We wanted to explore the effect of the time step $\Delta t$ in greater detail. The results confirm that longer evaluation time periods improve the accuracy of the forecasts, with the two week configurations always yielding improved accuracy. However, there seems to be a limit to this, since the longest time step of four weeks shows a clear increase in the error. We speculate that an optimal analysis window of around two weeks

exists for this watershed and that longer time frames would outlive the effective soil moisture memory span by assigning less importance to the most current history than it is rightfully due (in favour of older, less-relevant events). This time, with all configurations using both optimization objectives, we can see that there are no clear disadvantages of using optimization algorithms (but also no advantages). Experiment 3 also shows that the effect of the greed parameter $g$ is not very significant. That is, selecting some particles from dominated fronts to construct the target state distribution, and not only from the Pareto

front, does not seem to affect the results.

## 4 Conclusions and future work

In this article we introduced OPTIMISTS, a hybrid data assimilation algorithm that takes elements from sequential and variational methods in an attempt to harness the advantages of both of these inverse modelling perspectives. We conducted a set of streamflow forecasting factorial experiments on two watershed models of different resolutions—the Blue River with

1,105 state variables and the Indiantown Run with 33,455—using two different modelling engines: VIC and the DHSVM, respectively. Our results from assimilating streamflow data show that variational approaches are less prone to overfitting due to their extended analysis time window. On the other hand, aside from allowing probabilistic estimations of the state variables, sequential methods benefit from using Bayesian belief propagation because the knowledge encoded in the prior distribution helps maintain dynamical consistency. OPTIMISTS can be configured to take advantage of both of these

strategies. With our proposed sampling method, we also found that using optimization algorithms—as in sequential approaches— to create additional samples did not contribute to improving the accuracy of forecasts.

Aside from selecting which features from the traditional approaches are the most advantageous, our analysis demonstrated the benefits of judging candidate initial state assignments, not only for their ability to reduce the observational error but also, for their consistency with the model's physics. This multi-objective formulation proved essential for the high-dimensional

watershed, as it allowed the model to overcome the overfitting risk generated by the enlarged effect of equifinality. While a formal comparison with 4DEnVar (Buehner et al., 2010) or a similar hybrid method deserves an investigation on its own, this feature adds to a set of characteristics that makes our approach superior, at least conceptually: dominance-based sorting offers better exploration and balancing possibilities over those permitted by lumped cost functions, and the limitation to linear fitness functions is lifted here by using evolutionary algorithms.

However, the fact that OPTIMISTS actually reduced the forecast accuracy in one of the studied scenarios still suggests that better strategies can be implemented to enhance the effectiveness of data assimilation. For one, self-adaptive mechanisms (Karafotias et al., 2014) can be employed to select the assimilator's parameters that better match the specific conditions of a



forecast scenario—e.g., setting the assimilation time step as a function of the watershed's response time and the nature of the recent precipitation conditions. More importantly, initial state estimation should not be overburdened with the responsibility of compensating structural and parametrical deficiencies in the model. In this sense, we embrace the vision of a unified framework for the joint probabilistic estimation of structures, parameters, and state variables (Liu and Gupta, 2007), but we

remain sceptical of approaches that would increase the indeterminacy of the problem by adding unknowns without providing additional information or additional means of relating existing variables.

From a scalability perspective, we proposed simplifications to some computations in the OPTIMISTS algorithm to reduce the complexity of the memory and processing unit requirements. Together with the kernel density strategy used for state estimation, which proved to be efficient in describing the complex distributions with a relatively small ensemble size,

OPTIMISTS was able to improve the forecasting skill of the high-resolution model without significant computational overheads—running the model, even in parallel, accounted for more than 90% of the total assimilation time. However, our experiments also showed the benefits of considering the full set of dependencies between state variables—which had to be forfeited for the high-dimensional watershed. Future work should therefore explore means of allowing the efficient representation and exploitation of these dependencies as well.

**Acknowledgements**

This work was supported in part by the United States Department of Transportation through award #OASRTRS-14-H-PIT to the University of Pittsburgh.

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





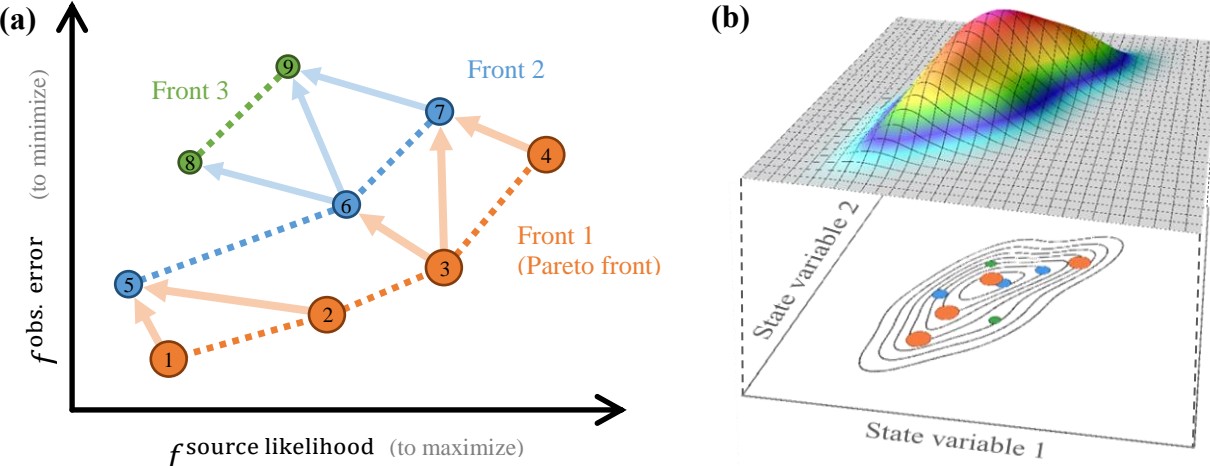

**Figure 1. (a) Nine labelled particles arranged into three fronts on a scatterplot of two filtering objectives (minimum observational error and maximum source likelihood). The dashed lines represent the fronts while the arrows denote domination relationships between particles in adjacent fronts. Particles in highly-ranked fronts are assigned larger weights $w_i$ (represented by particle size).**
5 **(b) Probability density (or likelihood) of an example two-dimensional state distribution constructed from the same nine particles using multivariate weighted kernel density estimation. Kernels centred on particles with higher weight (shown larger) have a higher probability density contribution.**





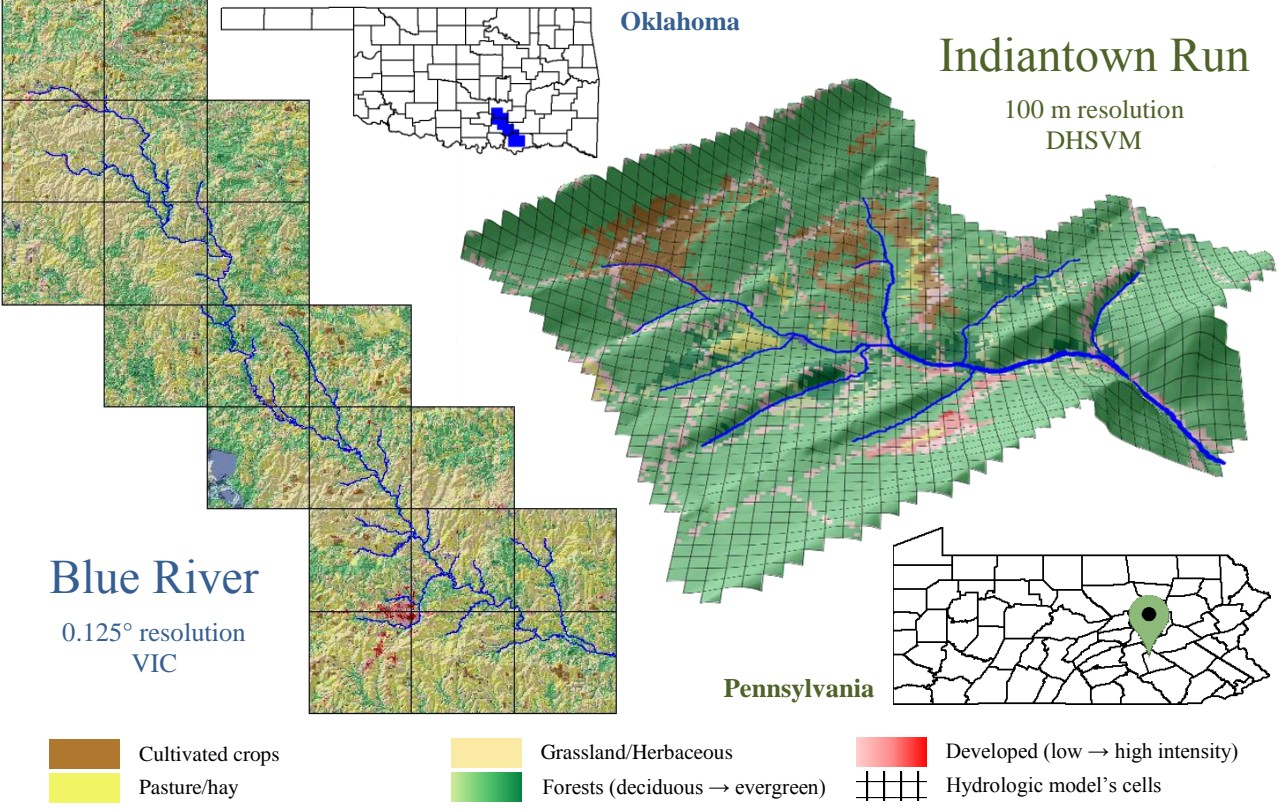

**Figure 2. Maps of the two test watersheds in the United States displaying the 30 m resolution land cover distribution from the NLCD (Homer et al., 2012). Left: Oklahoma's Blue River watershed 0.125° resolution VIC model (20 cells). Right: Pennsylvania's Indiantown Run watershed 100 m-resolution DHSVM model (1,472 cells).**





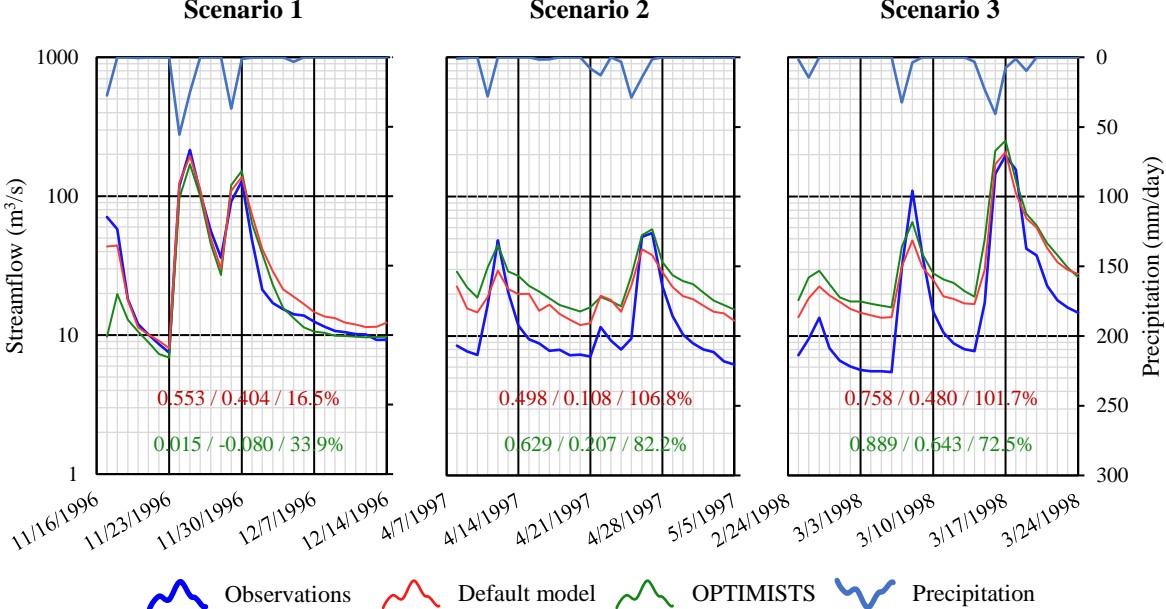

**Figure 3. Hydrograph comparisons for each of the three Blue River scenarios. The results from the default model and from the ensemble average of an OPTIMISTS configuration that showed good performance are compared with the observations from the stream gauge. In each scenario, the first two weeks correspond to the assimilation period while the latter correspond to the forecast period. The error metrics corresponding to the forecast period are indicated for the default model (above) and the OPTIMISTS ensemble (below) as follows: $NSE_{\ell 2}$ / $NSE_{\ell 1}$ / MARE.**



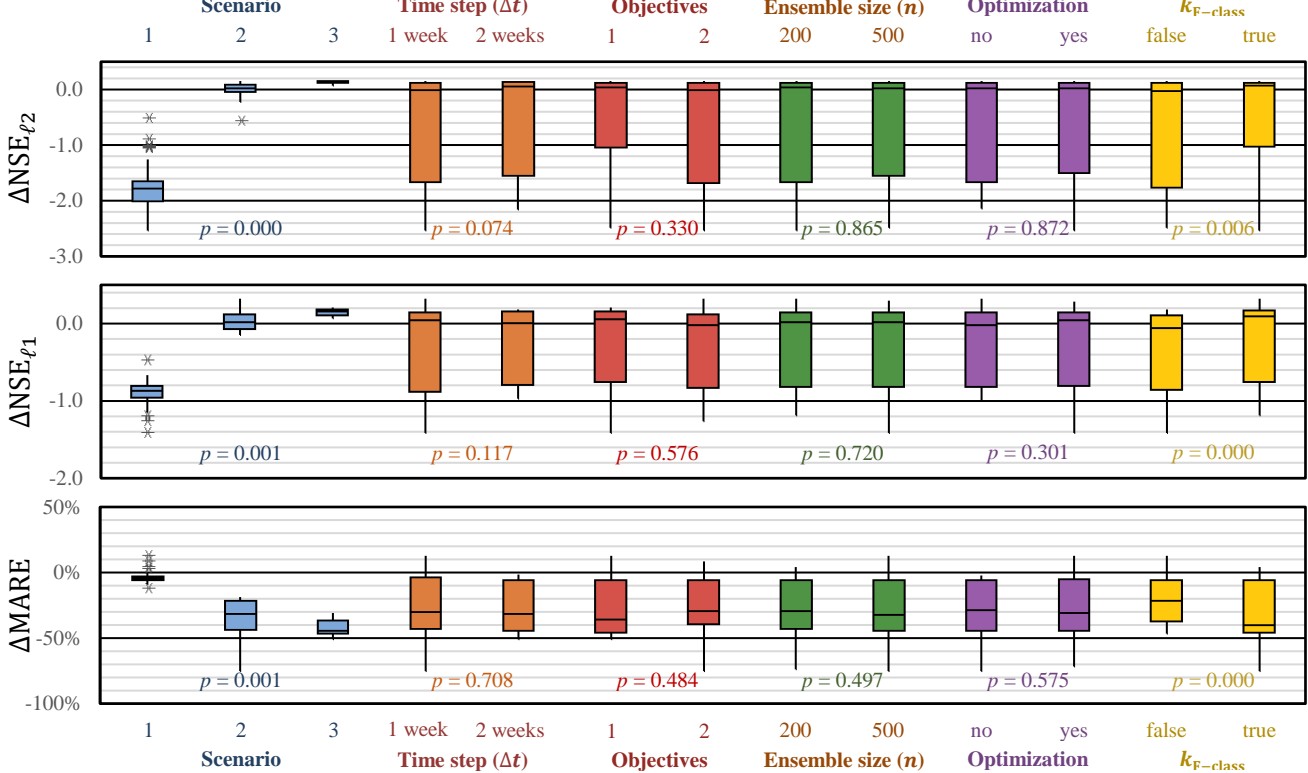

**Figure 4.** Boxplots of the forecasting improvements achieved while using OPTIMISTS on Experiment 1 (Blue River). Outliers are noted as stars. Each column corresponds to the distribution of the results on that scenario or assignment to the indicated parameter. For the $NSE_{\ell 2}$ and the $NSE_{\ell 1}$, values above zero indicate OPTIMISTS reduced the error, while values below zero indicate that it increased it. For the MARE, values below zero indicate error reductions. One objective means the particles' MAE was to be minimized; two indicate that also the likelihood was to be maximized. "no" optimization corresponds to $p_{samp} = 1.0$; "yes" corresponds to $p_{samp} = 0.4$. The $p$-values are determined using ANOVA (Montgomery, 2012), and indicate the probability that the differences in means between results with different parameter assignments are produced by chance (i.e., values close to zero indicate certainty that the parameter affects the results).





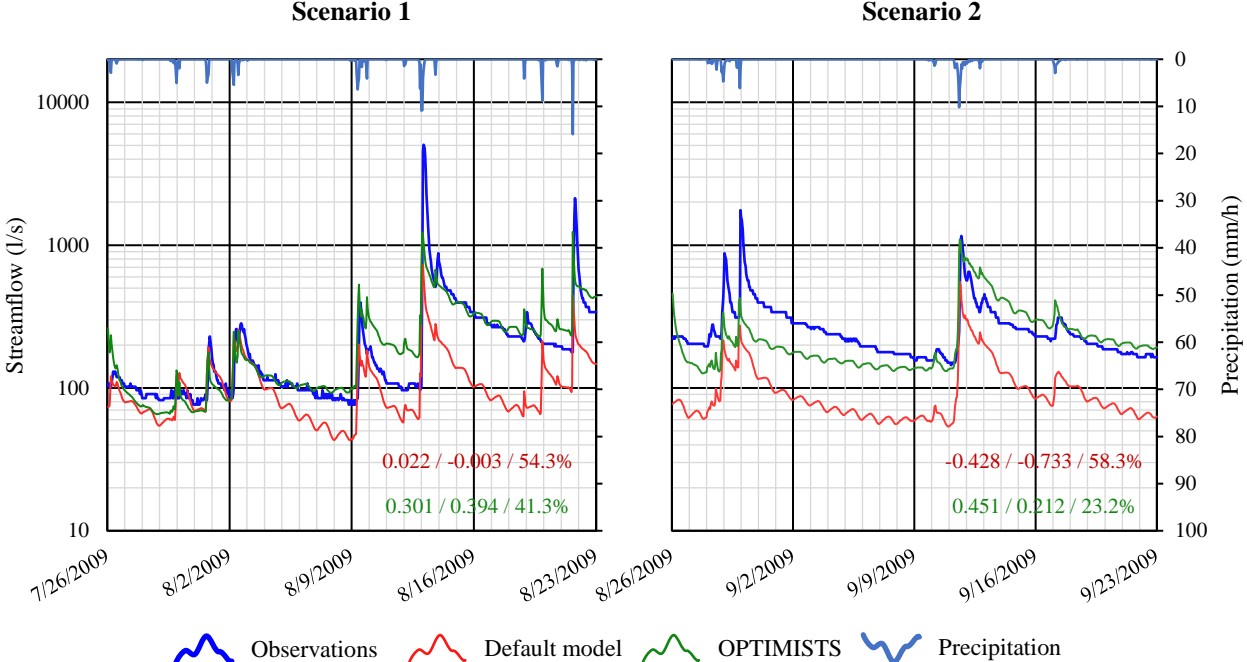

**Figure 5. Hydrograph comparisons for each of the two Indiantown Run scenarios. The results from the default model and from the ensemble average of an OPTIMISTS configuration that showed good performance are compared with the observations from the stream gauge. The first two weeks correspond to the assimilation period while the latter correspond to the forecast period. The error metrics corresponding to the forecast period are indicated for the default model (above) and the OPTIMISTS ensemble (below) as follows: $NSE_{\ell 2}$ / $NSE_{\ell 1}$ / MARE.**





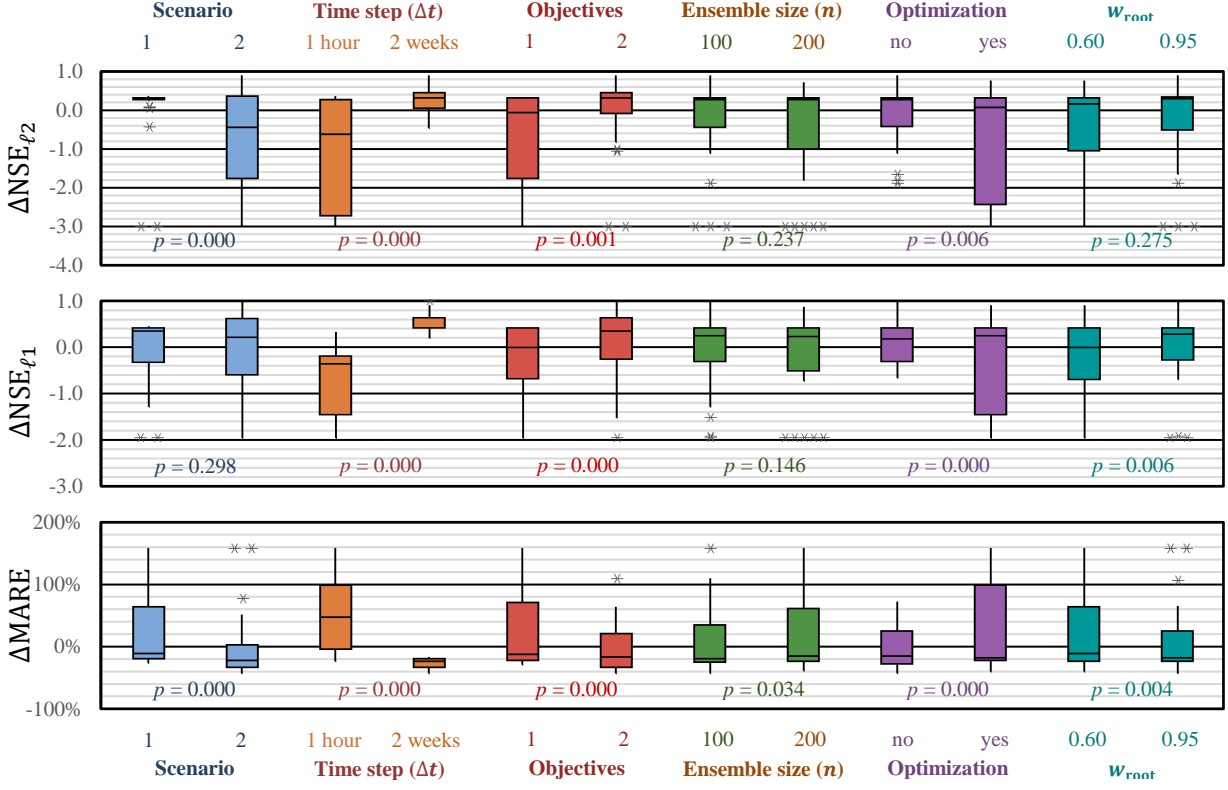

**Figure 6. Boxplots of the forecasting improvements achieved while using OPTIMISTS on Experiment 2 (Indiantown Run). Outliers are noted as stars and values were constrained to $NSE_{\ell 2} \geq -3$, $NSE_{\ell 1} \geq -3$, and $MARE \leq 200\%$. See the caption of Fig. 4 for more information.**





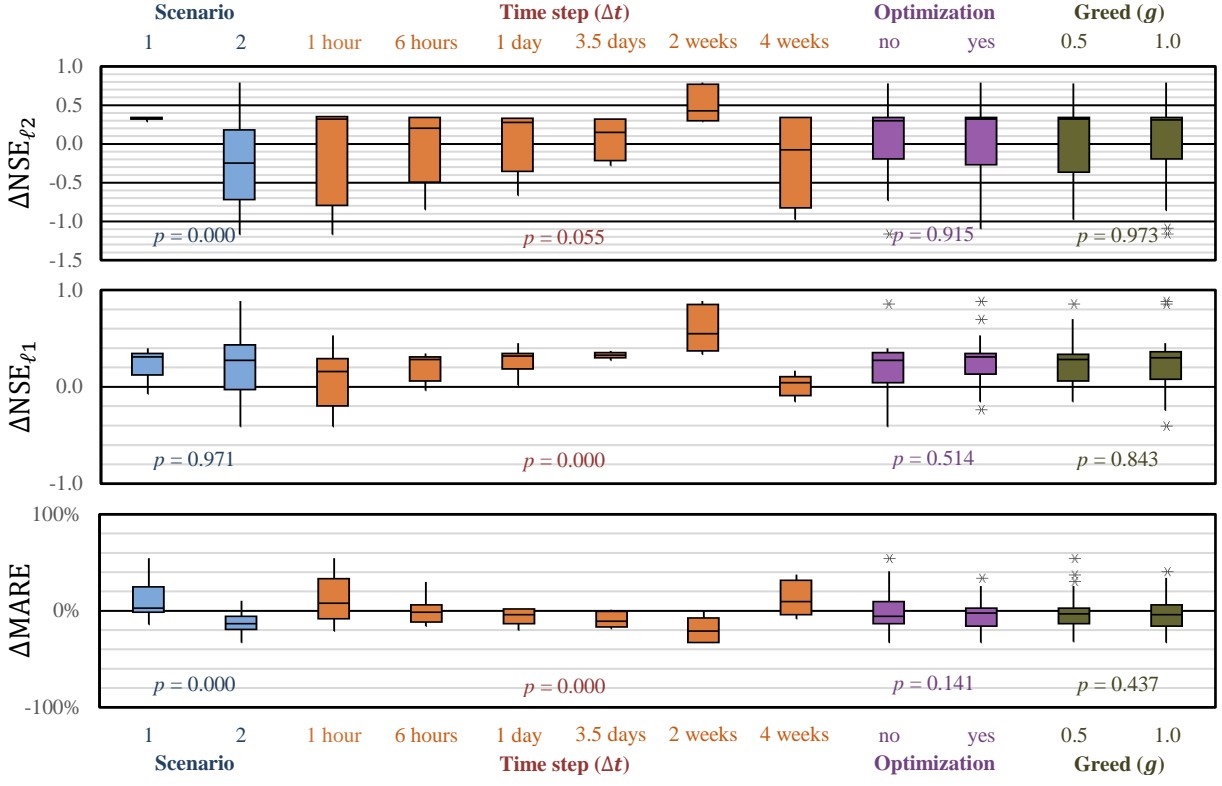

**Figure 7. Boxplots of the forecasting improvements achieved while using OPTIMISTS on Experiment 2 (Indiantown Run). Outliers are noted as stars. See the caption of Fig. 4 for more information.**





**Table 1. Comparison between the main features of standard sequential data assimilation algorithms (KF: Kalman Filter, EnKF: Ensemble KF, PF: Particle Filter), variational data assimilation (4-dimensional hard-constrained), and OPTIMISTS.**

|  | **Sequential** | **Variational** | **OPTIMISTS** |
|---|---|---|---|
| **Resulting state-variable estimate** | Probabilistic: Gaussian (KF, EnKF), Non-Gaussian (PF) | Deterministic (unless adjoint model is used) | Probabilistic (using kernel density estimation) |
| **Solution quality criteria** | High observational likelihood | Minimum observational error | Minimum observational error, maximum source state likelihood |
| **Analysis time step** | Same as model or observational time step | Entire data assimilation window | Flexible |
| **Search method** | Iterative Bayesian belief propagation | Convex optimization | Coupled belief propagation/multi-objective optimization |
| **Model dynamics** | Linear (KF), non-linear (EnKF, PF) | Linearized to obtain convex solution space | Non-linear (non-convex solution space) |

**Table 2. List of global parameters in OPTIMISTS**

| Symbol | Description | Range |
|---|---|---|
| $\Delta t$ | Assimilation time step (particle evaluation time frame) | $\mathbb{R}^+$ |
| $n$ | Total number of root states $\boldsymbol{s}_i$ in the probability distributions | $\mathbb{N} \geq 2$ |
| $n_{\text{pop}}$ | Population size for evolutionary algorithms | $\mathbb{N} \geq 2$ |
| $w_{\text{root}}$ | Total weight to allocate to drawn samples | $\mathbb{R} \in [0, 1]$ |
| $p_{\text{samp}}$ | Percentage of $n$ corresponding to drawn and random samples | $\mathbb{R} \in [0, 1]$ |
| $k_{\text{F-class}}$ | Whether to use F-class kernels or D-class kernels | true or false |
| $g$ | Level of greed for the assignment of particle weights $w_i$ | $\mathbb{R} \in [-1, 1]$ |



**Table 3. Characteristics of the two test watershed models: Blue River and Indiantown Run. US hydrologic units are defined in (Seaber et al., 1987). Elevation information was obtained from the Shuttle Radar Topography Mission (Rodríguez et al., 2006); land cover and impervious percentage from the National Land Cover Database (Homer et al., 2012); soil type from CONUS-SOIL (Miller and White, 1998); and precipitation, evapotranspiration, and temperature from NLDAS-2 (Cosgrove et al., 2003).**

| Model characteristic | Blue River | Indiantown Run |
|---|---|---|
| USGS station; US hydrologic unit | 07332500; 11140102 | 01572950; 02050305 |
| Area (km$^2$); impervious | 3,031; 8.05% | 14.78; 0.83% |
| Elevation range; average slope | 143 m – 406 m; 3.5% | 153 m – 412 m; 14.5% |
| Land cover | 43% grassland, 28% forest, 21% pasture/hay | 74.6% deciduous forest |
| Soil type | Clay loam (26.4%), clay (24.8%), sandy loam (20.26%) | Silt loam (51%), sandy loam (49%) |
| Avg. streamflow (90% range) | 9.06 m$^3$/s (0.59 m$^3$/s – 44.71 m$^3$/s) | 300 l/s (35 l/s – 793 l/s) |
| Avg. precipitation; avg. ET | 1,086 mm/year; 748 mm/year | 1,176 mm/year; 528 mm/year |
| Avg. temperature (90% range) | 17.26°C (2.5°C – 31°C) | 10.9°C (-3.5°C – 24°C) |
| Model cells; stream segments; $d$ | 20; 10; 1,105 | 1,472; 21; 33,455 |
| Resolution | 0.125°; daily | 100 m; hourly |
| Calibration | VIC's SCE built-in optimizer; 8 parameters; 85 months; objective: NSE$_{\ell_2}$ | OPTIMISTS' optimizer; 18 parameters; 20 months; objectives: NSE$_{\ell_2}$, MARE, absolute bias |

**Table 4. Setup of the three factorial experiments, including the watershed, the total number of configurations (conf.), the values assigned to OPTIMISTS' parameters, and which objectives were used (one objective: minimize observational MAE; two objectives: minimize observational MAE and maximize likelihood of source states).**

| No. | Watershed | Conf. | Parameters; objectives |
|---|---|---|---|
| 1 | Blue River | 32 | $\Delta t \in \{1\,\text{w}, 2\,\text{w}\}$, $n \in \{200, 500\}$, $n_\text{pop} = 25$, $w_\text{root} = 0.95$, $p_\text{samp} \in \{0.4, 1.0\}$, $k_\text{F-class} \in \{\text{false}, \text{true}\}$, $g = 0.75$; one or two objectives |
| 2 | Indiantown Run | 32 | $\Delta t \in \{1\,\text{h}, 2\,\text{w}\}$, $n \in \{100, 200\}$, $n_\text{pop} = 25$, $w_\text{root} \in \{0.60, 0.95\}$, $p_\text{samp} \in \{0.25, 1.0\}$, $k_\text{F-class} = \text{false}$, $g = 0.75$; one or two objectives |
| 3 | Indiantown Run | 24 | $\Delta t \in \{1\,\text{h}, 6\,\text{w}, 1\,\text{d}, 3.5\,\text{d}, 2\,\text{w}, 4\,\text{w}\}$, $n = 100$, $n_\text{pop} = 25$, $w_\text{root} = 0.95$, $p_\text{samp} \in \{0.4, 1.0\}$, $k_\text{F-class} = \text{false}$, $g \in \{0.5, 1.0\}$; two objectives |