# Peer review of "Hybridizing sequential and variational data assimilation for robust high-resolution hydrologic forecasting"

_Hydrology and Earth System Sciences, 2016_

## Referee Comment (RC1) · Anonymous Referee #1 · 25 Oct 2016

This study presents a hybrid approach to data assimilation by combining parts of sequential and variational algorithms, and demonstrate its applicability to hydrologic forecasting. The topic is interesting and appropriate for the journal. However there are certain problems with both the experiment design and the results. The design of the experiment is a bit confusing, no explanation si really given on the choice of the parameters etc., while the "scenarios" are not really explained well. In many of the comparisons the proposed algorithm degrades the performance when compared with the "default" model, which begs the question on why that is. Would a simpler data assimilation technique yield the same results? Given the increased complexity of implementing the OPTIMISTS algorithm compared to say an EnKF, I believe including the results

from implementing a simpler assimilation algorithm should be included. In addition, the description of the algorithm is somewhat confusing and doesn't answer the basic question of how this algorithm addresses the limitations of the sequential and variational approaches. Overall, I'm not convinced by the presentation or the results that this method can be superior to other data assimilation approaches as is stated in the Conclusions section.

p. 1, l. 22, "growing complexity": perhaps add a reference. p. 2, l. 30: what are these disadvantages? p. 3: does an observation need to be available for each assimilation step (seems so from l. 8)? In that case, wouldn't the assimilation step be limited by the observation time step? I suggest rewording l. 5 to clarify that. p. 3, l. 15: state variables can be output values too. I think using the term "predicted measurements" would be more appropriate. I have to admit I was confused by Section 2. The proposed algorithm seems like a mishmash of ideas from other algorithms, and there is no clear explanation what limitation the authors were trying to address by each choice they made on their algorithm. For example, why is it more advantageous to generate random samples to supplement the root samples with? Couldn't Steps 2-5 be replaced with one of the evolutionary algorithms that the authors are already using? What led the authors in choosing a complicated optimization algorithm that combines a GA and Metropolis-Hastings sampling instead of something simpler? Why isn't resampling adequate to account for sample impoverishment? I strongly suggest the authors rework the entire section by splitting the algorithm into its components and providing an explanation under each step describing what it does, what was the issue with current state-of-the-art approaches and how their proposed algorithm solves the issues. It would probably be useful to add a flow diagram describing the algorithm. With all the approximations made to make the algorithm computationally tractable, what is the advantage gained in comparison to simpler assimilation approaches? p. 8, l. 29: it's not clear why the experiment were configured the way they were. What was the rationale behind the choices in the parameters? For example, w_root is set to either a value or a range of values, but nothing really has been said about its significance or the possible

values it can take. When its value is within a range, did those values represent one of the factors in the factorial experiments? What were the discrete values used, if that is the case? Also, it might be good to reformat the table and use vertical lines to partition the columns. p. 8, l. 21 "To assess the performance of OPTIMISTS, three forecasting scenarios were selected...": what are these scenarios? p. 9, l. 1-2: were the meteorological forecasts sampled from a long-term climatology, i.e. ESP? p. 9, l. 4: this should be Section 4. p. 9, l. 13: why not add a table with the summary performance metrics? p. 9, l. 28-29: how much was the error reduced during the training period? p. 10, l. 6-8, "If these results cannot be generalized to challenge this entrenched assumption, they at least indicate that OPTIMISTS can efficiently encode the state probability distribution of complex models using relatively few particles—which directly translates to better scalability to higher-dimension applications.": I don't think that's necessarily true, since this assumes that there is no dependence on the sampling strategy to estimate the PDF. I suggest either augmenting this to strengthen the statement or restating. p. 11, l. 5: what is the ACF of soil moisture? p. 11, l. 22-24, "our analysis demonstrated the benefits of judging candidate initial state assignments, not only for their ability to reduce the observational error but also, for their consistency with the model's physics": wouldn't that be moot if the state was always generated by the model? p. 11, l. 25-27 "While a formal comparison with 4DEnVar (Buehner et al., 2010) or a similar hybrid method deserves an investigation on its own, this feature adds to a set of characteristics that makes our approach superior, at least conceptually": I'm having a hard time with this statement, especially given the results and the lack of any comparison with other DA approaches. p. 11, l. 30: unless I'm missing something, the proposed method reduced the forecast accuracy in a number of cases and not just "one of the studies scenarios".

---

## Referee Comment (RC2) · Anonymous Referee #2 · 1 Nov 2016

HESS-Discussions

DOI: 10.5194/hess-2016-454

Title: Hybridizing sequential and variational data assimilation for robust high-resolution hydrologic forecasting

Authors: Felipe Hernandez and Xu Liang

Review

The paper describes a "hybrid" algorithm for streamflow data assimilation. The algorithm, termed OPTIMISTS, mixes and matches elements from ensemble-based and

variational concepts. A number of assimilation tests are conducted for two watersheds, the first with ∼1,100 state variables and the second with ∼33,000 state variables. These tests assimilate streamflow data for 2 weeks or 4 weeks and then conduct streamflow "forecasts" using the final time of the assimilation period as initial condition. The skill of the forecasts is compared to a control forecast for which the initial condition does not benefit from the streamflow assimilation.

The authors claim that their new "hybrid" algorithm is superior, at least conceptually, to other hybrid approaches. They acknowledge that forecasts based on streamflow assimilation using the new algorithm are in some cases worse than those without.

The topic of the paper is appropriate for HESS. However, my summary assessment is that the paper is unusually difficult to follow and lacking in specific, clear explanations and results. Most importantly, it is never quite clear what motivates the choices made in constructing the new algorithm and how (and by how much) they are better than the corresponding elements from existing algorithms that the authors reject. The tests only show the difference between the new assimilation algorithm and the default (calibrated) model. But since the authors are touting their new algorithm as an improvement over other hybrid algorithms, they need to show that their algorithm outperforms the others. But I am doubtful that this would be possible, given that, on balance, the new algorithm does not seem to outperform the model-only results.

Overall, I do not think that the paper is suitable for publication in HESS at this time.

Major comments:

1) By far my greatest concerns is that the tests in the results section only show the difference between the new assimilation algorithm and the default (calibrated) model. But since the authors are touting their new algorithm as an improvement over other hybrid algorithms, they need to show that their algorithm outperforms the others, or at least is capable of improving the model-only results (which, on balance, is not the case). The statement (p11, L26) that "a formal comparison with 4DEnVar (Buehner et al 2010) or

a similar hybrid method deserves an investigation on its own" is woefully inadequate. It is important to keep in mind that the authors put together the new algorithm by picking and choosing elements from existing concepts at will, using "features that [the authors] consider most valuable" (p2, L32). The proposed new algorithm as a whole has not been proven elsewhere (as is the case for standard algorithms if the application satisfies the usual conditions of linearity and error independence and whiteness). Put differently, is the new algorithm optimal under those assumptions?

2) It is extremely difficult to follow the methods section. The authors simply state what they do (and sometimes more, see next comment) without clearly explaining what it is that motivates a particular choice and why this choice is likely better than what the existing algorithms do. For example, step 2 in the 7-step list on p3 (re. random samples) replaces the resampling in traditional particle filter algorithms (p4, L4). But why should this new approach better "avoid sample impoverishment" (p4, L4) than the existing resampling strategies? Note that I am not saying that it does not. My comment is that it is not sufficiently motivated, justified, and explained.

3) Sections 2.1 and 2.2 include three different approaches for the likelihood (eq. 3, 7, and 8), each representing a consecutively more aggressive simplification. But it looks like only eq. 7 and 8 were used in the tests (p8, L31-32). Why is equation 5 needed? It seems that the discussion lacks focus.

4) Abstract: The text here is far too generic and not reflecting the results of the assimilation tests. p1, L17-18 talks about "the benefits of the coupled probabilistic/multi-objective approach", but quantitative statements are missing (how much improvement?), and the abstract does not include any hint re. the fact that, on balance, the tests do NOT show an improvement from the assimilation compared to the model-only results.

5) The text includes a lot of optimization and particle filter jargon, but I am not sure that it is always used correctly. I am not an expert in particle filtering per se, but I am

convinced that breaking data assimilation algorithms down into "sequential" and "variational" is not a good way to approach the topic (p1, L7; p2, L4-5). There are variational algorithms (3DVAR, PSAS) that are sequential. Also, the Kalman filter update equation can be derived from the same objective function that is usually considered the starting point for the derivation of variational algorithms such as 3DVAR (that is, under certain assumptions 3DVAR and the Kalman filter are just different ways of solving the same problem). Finally, there are non-sequential (ensemble) Kalman smoother algorithms that are based on Bayesian principles and cover an assimilation window of non-zero length. How would the authors classify these Kalman smoothers? The point here is that the sloppy use by the authors of the most basic terminology raises suspicions that the more arcane details of the particle filter and optimization language are equally inaccurate, which casts doubts on the validity of the entire development.

6) Another example of an odd statement is on p1, L11: "...which promotes the reduction of observational errors..." I am guessing by this statement the authors refer to the reduction of the errors in the *analysis* or forecast estimates resulting from the data assimilation. The "observational errors" themselves cannot be reduced by data assimilation. Moreover, all assimilation algorithms promote the reduction of the error in the *analysis* estimates. This is not just the case for the algorithm proposed by the authors. Besides being sloppy, the statement therefore also provides literally no useful information.

7) Another sloppy statement is on p1, L21: "...geophysical models are as underdetermined as ever...". (also p1, L7-8) I am guessing that the authors mean that model *estimates* of parameters or states based on the *assimilation* of (relatively few) observations are underdetermined. The geophysical models themselves are not underdetermined. The optimization (or assimilation) problem is.

8) Section 1 (Introduction): The text is missing focus on hydrological data assimilation. Only the title reveals that the introduction is really about estimating streamflow, while references are a mix of hydrological assimilation and atmospheric or ocean assimilation. While some features of the assimilation or optimization are shared across disciplines, an algorithm designed for an NWP system cannot just be applied to hydrological models without consideration of the fundamental differences in the two kinds of models. E.g., processes in GCMs are chaotic, but hydrological models are about damped physics. The dimensionality of hydrological models is far smaller than that of GCMs used on modern NWP systems. The point is that the Introduction could do a much better job of introducing the research and results.

9) p8,L17: The state variables include interception? Is there any chance that the amount of water intercepted by the canopy can be adequately estimated from the assimilation of streamflow observations? I would venture to guess that the estimated interception reservoirs are simply noise.

10) p8, L21-22 mentions three forecasting "Scenarios" but I could not figure out what exactly those are. There are two watersheds and 3 Experiments, each with a larger number of "factorial" experiments. But what do the authors mean by "Scenario"? Lack of this information (or its prominent exposition) makes it very difficult to understand the results.

11) p9, L25-26: "For Experiment 1, the correlation between the improvement of the $NSEl2$ during the assimilation period and the improvement during the forecast period was of -0.344, and of -0.669 for the $NSEl1$." I did not understand this sentence at all.

12) p10, L4: *Why* could it be that the performance is indifferent to n (number of particles or ensemble members?)

13) p11, L27-30: So why does the "conceptually superior" new algorithm (p11, L27) give such poor results of the test runs (p11, L30)? The authors do not provide an adequate explanation. Stating how the algorithm could be improved (as the they do in the subsequent lines) is not sufficient. At best, the new algorithm can still be improved, which needs to be implemented and shown. Thereafter it could be considered as a step forward. As things stand, the reader is left wondering whether there is a problem

with the new algorithm or whether the problem is ill-posed and standard assimilation algorithms would also fail.

Minor comments:

a) I am not sure whether this is a journal requirement, but it is odd to have the equations placed at the end of each paragraph. Typically, an equation is referenced but then shown only a few lines further down.

b) p9, L1-3: The "forecasts" use NLDAS-2 forcing data, which is equivalent to using perfect meteorological forecasts. Using the term "forecasts" throughout the manuscript is therefore somewhat misleading. The experiments are really more like "simulations" than "forecasts".

c) p9, L8: replace "valleys" with "periods between runoff peaks" or something similar? It is not obvious what is meant by "valleys".

d) p11, L1: Caption of Fig 7 says that the results are for Experiment 2, whereas the text here says they are for Experiment 3. Which is it?

---

## Author Comment (AC1) · 14 Dec 2016

Felipe Hernández and Xu Liang

feh17@pitt.edu

First we would like to thank Referee #1 and Referee #2 for their thorough and careful review of our manuscript, and for their very helpful comments. We will try our best to incorporate their suggestions, address their concern, and modify our manuscript accordingly when we are given the green light. We believe implementing their valuable comments/suggestions will improve the presentation, clearness, and accuracy of our work.

We agree that a comparison between OPTIMISTS and an established method would be highly desirable. We actually developed a Particle Filter (PF) with "regularized" re-sampling using a Gaussian kernel for this purpose, but were unable to run VIC reliably

for a single time step (we only managed to allow saving and recovering the routing state variables for simulations of at least a couple of days). This is due to a problem in our coupled version between VIC and the routing scheme, which will require additional efforts to be fixed. On the other hand, comparisons using the DHSVM model would not work because traditional PFs are not suited for high-dimensional models (this is precisely one of the problems we are addressing with OPTIMISTS) and, similarly, the DHSVM is not designed for modeling with coarse-resolutions.

Moreover, we had an additional reason why we were not totally convinced about including a comparison in the first version of the manuscript. OPTIMISTS is as much a new data assimilation (DA) method as it is a hybrid of existing methods. For example, it can be configured to behave as an evolutionary 3D-Var algorithm (select a sequential time step, evaluate many solutions with a single cost-function objective to be minimized, and do not use any root samples). Enlarging the assimilation time step transforms it into a type of 4D-Var. By using only root and random samples, a sequential time step, and an objective that evaluates the likelihood of candidate solutions given the observations, one can get something similar to a PF. Therefore, to some extent, the comparisons we performed between multiple configurations of OPTIMISTS could represent comparisons between those methods. We are making modifications to the manuscript to better convey this idea.

This flexibility in OPTIMISTS can be very advantageous, as it allows one to find configurations that better match specific conditions. However, as can be seen in the experiments, it comes at the cost of not allowing to consistently get adequate results if it is not well parameterized. Part of the objective of the experiments was to determine which components from the methods that OPTIMISTS borrows from are the most beneficial. It is very clear, for example, that sequential DA performs poorly in our tests, suggesting that traditional EnKF, PF, 3D-Var, etc. would most likely underperform compared to 4D-Var or some other method with time-extended evaluations. Similarly, one should configure OPTIMISTS to take advantage of the extended-period evaluations.

From Figures 6 and 7 it is clear that selecting the adequate time step will almost guarantee that the method will yield improved forecasts. On the other hand, choosing the sequential route will result in very poor performance. We will modify the manuscript to better convey the idea that, after selecting adequate parameters (extended assimilation time frame, multiple objectives), OPTIMISTS should, on balance, provide good results. In other words, it is not the entire range of possible configurations of OPTIMISTS that results in better forecasts, but a subset of them which the user should adapt based on their specific watersheds. The information shown in our results would help identify a good starting point for this adaptation.

Nonetheless, we used this opportunity to implement a 4D-Var algorithm and we performed comparisons on the VIC model. This implementation of 4D-Var uses a single-objective version of the ensemble optimization algorithm in MAESTRO and, given its evolutionary nature, is able to solve the non-linear optimization problem. A traditional two-term cost function (normalized squared deviations from observations and from the background) was used as the objective to be minimized. It must also be noted that, to speed convergence, we seeded the initial population with the root/base states (as it is done for OPTIMISTS).

Before discussing the comparison results we want to clarify the nomenclature of our experiments, which we will also improve in the manuscript during the revision process. The "scenarios" correspond to four-week time frames in which we evaluate the models, with a two-week assimilation period and a two-week forecasting period. We have three scenarios for the Blue River VIC model started in the following dates: 11/16/96, 4/7/97, and 2/24/98. Similarly, we have two scenarios for the Indiantown Run DHSVM model starting in 7/28/09 and 8/26/09.

The attached figure shows the comparison between one of the best-performing configurations of 4D-Var (using 2,000 iterations) and OPTIMISTS (1-week period, 500 members, 2 objectives, 40% samples, 95% roots, K-class kernels, and a greed value of 0.75). The numbers in the figure correspond to the three error metrics (NSEI2, NSEI1,

MARE) during the two forecast weeks. This comparison shows that, even though 4D-Var outperforms OPTIMISTS in Scenario 1, its forecasting accuracy is still inferior to the default model. This gives validity to the explanation in the manuscript, according to which the special circumstances of the scenario are unfavorable (no significant precipitation occurs during that period after a large storm) for DA in general. This result shows that it is not necessarily a sign of a weakness particular to OPTIMISTS. In addition, 4D-Var also outperforms OPTIMISTS in Scenario 2, but OPTIMISTS gets the upper hand on Scenario 3.

While the results with 4D-Var are in general slightly better than those of OPTIMISTS, our approach does have important redeeming qualities. First, it requires a considerable smaller number of evaluations to reach results that are comparable or even better than those of 4D-Var. Second, it provides a probabilistic estimate. And third, it leverages the inclusion of the Bayesian approach to mitigate the exponential growth of the solution space for cases of higher dimensionality. We will include this comparison in the manuscript to provide the readers with a more complete picture on what performance one can expect from OPTIMISTS in contrast to state of the art methods. Of course, as we mentioned, more systematic experiments should be carried out to better establish the benefits and weaknesses of the algorithm.

On another topic, we especially appreciate Referee #2 for pointing out the problems related to the broad classification of DA methods we used (i.e., sequential and variational). Even though we have seen this very distinction being used in the literature before, it is a very good point that these two categories are not sufficiently differentiated. However, despite the misleading labels, we believe that the distinction we were illustrating was between Bayesian (KF, EnKF, PF) and variational (1D-4D) methods (a fair one, given that the key difference is the actual method for exploring the solution space). We will both adopt these more accurate labels in the revised manuscript (including the title) and acknowledge that even these modified categories do not allow for a perfect separator for the wealth of methods that exist–they are more a practical

framework from which to build a productive discussion. After all, one of our objectives with OPTIMISTS is to build bridges between these approaches.

We similarly appreciate the referees for identifying inaccurate uses of technical terms. We will fix these problems in the revision process and attempt a better integration of the terminologies typically used in the multiple fields that make up the intersection where our work stands, including Kalman filtering, particle filtering, variational DA, and evolutionary computation.

Response to other comments:

- We will gladly revise the introduction and the method sections to make them serve the audience of the journal better. In the original manuscript, we intentionally made the introduction and the methods sections relatively domain-agnostic, with the hope that our approach could be applied beyond our own domain of Hydrology. We understood that this came at the cost of denying hydrologists a better degree of familiarity throughout the explanation, but we had hoped it would open the possibility of furthering the dialogue between additional disciplines that use DA and it may be worth the cost. But as the reviewer pointed out, the sacrifice of the clarity and concise presentation needs to be considered. In the revision, we'll try to keep a good balance and make the clarity the first priority.

- We decided to include the full array of state variables that both models (VIC and DHSVM) possess for our tests, even those which would indeed just represent noise in the large scale of things (such as the interception). This is because we want to test the robustness of OPTIMISTS. A robust DA algorithm should be able to manage the inclusion of such type of variables (especially under the threat of equifinality), while omitting these variables would help simplifying the problem. Although it would make practical sense to focus on the most important state variables, their inclusion in our tests effectively represent an additional handicap on the method, increasing the significance of any positive results.

- We use an ensemble optimization algorithm because of its proved advantages from the evolutionary optimization literature (in essence, by trying to mitigate the problems associated with the no-free-lunch theorems). Yes, one could use a simpler algorithm (like the standard NSGA-2).

- While we did not use Eqs. 3, 4, and 5 in our tests, these would provide the most accuracy for models with low dimensionality. We have them included for the sake of completeness and for the applicability of our method to be easily extended to problems with a low dimensionality. In the revised manuscript we will make this point clear.

We will also make improvements in response to other comments, including:

- Dividing the explanation of the method into sub-sub-sections to allow for an easier understanding

- Adding reasons for design decisions of the algorithm and their contrasts with existing methods where they are currently missing

- Better explaining the difference between using random samples and using traditional resampling

- Further discussing the differences in the availability of meteorological information in our tests and in a hypothetical operational case

- Reformatting of Table 4 to improve readability (Values in brackets are not ranges but rather a set: these were all the levels assigned to the factors/parameters in the factorial experiment.)

- Adding a table with a summary of the performance metrics of the multiple configurations We would like to thank the referees again for their valuable comments and suggestions. With the approval of the editor, we will work on the aforementioned modifications to revise the manuscript and will be glad to implement additional changes to meet the journal's and the referees' standards.

[Figure]

[Figure]

**Fig. 1.** Comparison between OPTIMISTS and 4D-Var on the Blue River model